# Significance of Non-Statistically Significant Results in the Prediction of Oral Health-Related Quality of Life in Orthodontic Patients: A Survey Using Two Linking Questionnaires

**DOI:** 10.3390/ijerph20085446

**Published:** 2023-04-10

**Authors:** Peerapong Santiwong, Kantrakorn Dutsadeeviroj, Kittithad Potchanarungvakul, Thanpat Leartchotikul, Thanachot Jiwsiritrakul, Kawin Sipiyaruk

**Affiliations:** 1Department of Orthodontics, Faculty of Dentistry, Mahidol University, Bangkok 10400, Thailand; 2Doctor of Dental Surgery Program, Faculty of Dentistry, Mahidol University, Bangkok 10400, Thailand

**Keywords:** dietary difficulty, dental education, oral health-related quality of life, orthodontic practice, patient-centered care

## Abstract

Patients with a fixed orthodontic appliance may have difficulties in maintaining good oral health-related quality of life (OHRQoL), and evaluating self-perceived OHRQoL in orthodontic patients could be challenging for their orthodontists. Therefore, this research was conducted to investigate whether orthodontic postgraduates would accurately evaluate the OHRQoL of their patients. Two self-administered questionnaires were designed for patients to rate their OHRQoL and for their orthodontic postgraduates to evaluate them on OHRQoL. All pairs of patients and their orthodontic postgraduates were requested to independently complete the questionnaires. Pearson’s correlation and multiple linear regression were performed to determine the relationships of the variables and to identify significant predictors on OHRQoL, respectively. There were 132 pairs of orthodontic patients and their residents who completed the questionnaires. There were no significant correlations between OHRQoL perceived by patients and evaluated by their orthodontic postgraduates in all aspects of treatment needs and dietary difficulties (*p* > 0.05). In addition, the regression model demonstrated no significant predictors for the self-perceived treatment needs and dietary difficulties of orthodontic patients. There seemed to be challenges for orthodontic postgraduates to evaluate their patients’ oral health-related quality of life. Therefore, the OHRQoL measures should be increasingly implemented in orthodontic education and practice in order to enhance the concept of patient-centered care.

## 1. Introduction

Malocclusion and jaw deformities can lead not only to self-perceived disfigurement but also difficulties in chewing food [1]. Orthodontic treatment can correct the tooth alignment of patients, leading to the improvement of occlusion and appearance [2]. As a result, both the physical and psychological well-being of patients can be improved. To improve the tooth alignment with fixed orthodontic treatment, brackets and archwires are commonly used for tooth movement. These attachments and equipment might cause difficulties during the treatment, and negatively impact the oral health-related quality of life (OHRQoL) of patients.

Although orthodontic treatment can improve the quality of life of patients, there could be some difficulties during the treatment procedure such as dietary limitations, oral hygiene problems, and pain [3,4,5]. Patients with fixed orthodontic appliances could feel pain when chewing food [6] and are likely to feel annoyed with food impaction in orthodontic brackets or archwires [7]. Social disability can also be found during orthodontic treatment, as patients may feel embarrassed to smile or to have a meal in public [3]. These challenges may inhibit patients to maintain their emotional stability, as they cannot handle their routine tasks as usual [8,9]. They can negatively affect physical and psychological well-being, leading to a decrease in patient compliance during the orthodontic treatment. 

These difficulties during orthodontic treatment are important issues of concern, however, it appears to be quite challenging for orthodontic postgraduates to evaluate these difficulties in patients as each patient seems to perceive these difficulties differently [10]. Different types of fixed orthodontic appliances and phases of treatment can also affect self-perceived OHRQoL [9,11]. Interestingly, self-perceived OHRQoL does not seem to be correlated with the severity of malocclusion [12,13]. In other words, it can be quite challenging to predict the OHRQoL of patients in orthodontic practice. However, the concept of patient-reported measurements can be implemented to collect the self-perception from orthodontic patients [14]. This would support orthodontic residents to provide a patient-centered care in their practice. 

Patient-reported measurements appear to be supportive in understanding OHRQoL including the dietary difficulties of patients, however, they have not been widely encouraged in orthodontic education. In addition, there is no clear evidence demonstrating that this approach is necessarily required for orthodontists to realize how their patients perceive their OHRQoL during the treatment. However, orthodontic residents who have few clinical experiences may have challenges in evaluating the OHRQoL of their patients. Consequently, the aim of this research was to investigate whether or not orthodontic postgraduates would accurately evaluate the self-perceived treatment needs and dietary difficulties of their patients. The findings of this research would enable orthodontic instructors and educators to realize the importance of patient-reported measurements to evaluate self-perceived OHRQoL in orthodontic education. 

## 2. Materials and Methods

### 2.1. Research Design

This research employed a quantitative design using two self-administered questionnaires as data collection tools. The first questionnaire was constructed for patients to rate their OHRQoL, while the second form was for their orthodontic postgraduates to evaluate them on malocclusion types and severity, treatment needs, and dietary difficulties. All pairs of patients and their orthodontic postgraduates were requested to independently complete their questionnaires. The patients were first recruited, and if they agreed to participate in this research, their orthodontic postgraduates would be requested to complete the questionnaire. As each postgraduate was required to provide orthodontic care for a number of patients, most of them were asked to complete the questionnaire a couple of times. The numbers of their questionnaire completion depended on the patients who agreed to participate in this research. The data retrieved from the two questionnaires were then analyzed to address the two research objectives. The data collection was conducted between November 2021 and May 2022 at the Orthodontic Clinic, Faculty of Dentistry, Mahidol University.

### 2.2. Research Participants

The population in this research was patients undergoing fixed orthodontic treatment at the Orthodontic Clinic, Faculty of Dentistry, Mahidol University. Patients were included if their age was 15 years old or above and if they received treatment by orthodontic residents. However, they were excluded if they had congenital craniofacial abnormalities, underwent removable appliance treatment, or were not able to read fluently in Thai. According to the sample size calculation for a finite population, to achieve the confidence level at 95% and a margin of error of 5%, a total of 132 patients were recruited to complete the questionnaires, who were receiving orthodontic care from 38 residents.

### 2.3. Questionnaire Design

The questionnaires for both patients and orthodontic postgraduates were constructed based on a review of the literature [4,5,15,16,17]. All items of both questionnaires were first constructed in English but afterward translated to the Thai language to facilitate the participants and minimize response bias with the evaluation of translation validity. 

The questionnaire for orthodontic patients (PT questionnaire) consisted of five sections (27 items): (1) Self-perceived needs of orthodontic treatment in functional limitations (five items); (2) Self-perceived needs of orthodontic treatment in esthetic problems (five items); (3) Self-perceived needs of orthodontic treatment in psychosocial concerns (five items); (4) Dietary difficulties in physiological (functional) limitations (six items); and (5) Dietary difficulties in psychological aspects (six items). All of these items were designed using five-point Likert scales, in which ‘1’ refers to ‘Strongly disagree’; ‘2’ refers to ‘Disagree’; ‘3’ refers to ‘Neither agree nor disagree’; ‘4’ refers to “Agree’; and ‘5’ refers to ‘Strongly agree’.

The questionnaire for orthodontic postgraduates (PG questionnaire) consisted of two sections, which were (1) the malocclusion information of patients (eight items) and (2) the professional evaluation on the treatment needs and dietary difficulties of patients (five items). The questions in the first part were designed using a single-select question (categorical data), while those in the second part were visual analog scales (0–100). The visual analog scales were selected for the PG questionnaire, rather than the constructs containing a number of items as the PT questionnaire, in order to minimize the burden for orthodontic residents who were required to complete the questionnaire a couple of times.

### 2.4. Questionnaire Reliability and Validity

Content validity was first performed following the completion of the questionnaire design. Two orthodontists and one advanced general dentist who were experts in OHRQoL were requested to evaluate and rate all questions of the two questionnaires (−1, 0, 1), and whether or not they could have addressed their objectives. The questionnaires were revised and evaluated repetitively until the Item-Objective Congruence IOC) scores of all items were 0.5 or above. In addition, to enhance the translation validity, the original version of the questionnaire was translated into Thai by the research team, and the back-translation was performed by an orthodontist who was fluent in English. The similarities and differences of the two English versions were compared and scored (−1, 0, 1) independently by three general dentists who were fluent in English to evaluate whether or not there was any point of translation to be improved. The translation was performed repetitively until the score of all questions was over 0.5. 

To enhance the reliability of both questionnaires, the validated versions were piloted and evaluated using the test–retest reliability and Cronbach’s alpha to ensure that they were reliable over time and that their constructs were appropriate in terms of internal consistency, respectively. The test–retest reliability was carried out on both questionnaires by asking 15 orthodontists and 30 orthodontic patients to complete the PG and PT questionnaires, respectively, twice with a two-week interval. The intraclass correlation coefficient (ICC) of 0.7 was expected for all items/constructs of both questionnaires to be acceptable. The Cronbach’s alpha was performed by using the data from the first pilot completion of the PT questionnaire, while the internal consistency was not required for the PG version, as it was not the measurement of the theoretical constructs. The coefficient alpha of 0.7 was expected for acceptable internal consistency. The problematic items of the questionnaires were revised or deleted until all questions and all constructs achieved the acceptable value (0.7). Following the iterative process of the questionnaire improvement, all items and constructs were quite reliable according to the data from the pilot testing (Table 1).

### 2.5. Data Analysis

The data collected was analyzed using SPSS (Statistical Package for the Social Sciences). An overview of the research data was presented using descriptive analysis. All data were normally distributed according to the Kolmogorov–Smirnov test. The relationships of each variable were determined using Pearson’s correlation. Multiple linear regression was used to identify whether or not the independent factors (‘Malocclusion information’, ‘Evaluated treatment needs’, and ‘Evaluated dietary difficulties’) had an effect on the dependent variables (‘Self-perceived treatment needs’ and ‘Self-perceived dietary difficulties’). Statistical significance was taken at *p* < 0.05.

### 2.6. Ethical Considerations

The data retrieved from the questionnaires were not purely anonymous, as the linkages between the two forms were required. However, the coding was implemented rather than the identifiable data of the participants to assure the confidentiality of the participants in order to minimize the response bias. Informed consent was obtained from all participants involved in the study. For the participants under the age of 18, the consent of their guardian was required in accordance with the ethical consideration. This research protocol was conducted in accordance with the Declaration of Helsinki, and approved by the Faculty of Dentistry and the Faculty of Pharmacy, Mahidol University, Institutional Review Board (MU-DT/PY-IRB), reference number: MU-DT/PY-IRB 2021/086.0610 on 6 October 2021.

## 3. Results

### 3.1. Internal Consistency of the PT Questionnaire

The data from the PT questionnaire retrieved from the sample pool of this research were analyzed to assure the internal consistency. The coefficient of Cronbach’s alpha demonstrated a high internal consistency of all constructs, as presented in Table 2.

### 3.2. Patient Information of Research Participants

There were 132 pairs of orthodontic patients and their residents who completed and returned the questionnaires. Of those 132 patients, 45 of them (34.09%) were male and 87 (65.91%) were female. Their average age was 23.01 (SD = 7.78), ranging from 15 to 31 years old. The descriptive statistics of patient information representing the types and severity of malocclusions are presented in Table 3. 

### 3.3. OHRQoL Perceived by Patients and Their Orthodontists

The orthodontic treatment needs and dietary difficulties during the treatment perceived by the patients themselves and evaluated by their orthodontic postgraduates are presented in Table 4. Esthetic aspects were rated as the largest issues for the need for orthodontic treatment by both patients (3.88 from 5) and the orthodontic postgraduates (80.60 from 100). Patients perceived that the functional limitations were the least important reasons for orthodontic treatment (2.84 from 5), while the postgraduates believed that they were psychosocial concerns (62.3 from 100). Both patients and postgraduates perceived that dietary difficulties during the treatment were physiological limitations, higher than those of psychological concerns. 

### 3.4. Correlations of OHRQoL Perceived by Patients and Evaluated by Orthodontic Postgraduates

There appeared to be no significant correlations between OHRQoL perceived by the patients and evaluated by their orthodontic postgraduates in all aspects of treatment needs and dietary difficulties during the treatment (*p* > 0.05), as demonstrated in Table 5. 

### 3.5. Influential Factors on Self-Perceived Treatment Needs of Orthodontic Patients

The regression model demonstrated that all independent variables appeared to have no statistically significant impact on the self-perceived treatment needs of orthodontic patients (*p* > 0.05) with an R-squared value of 0.063 (Table 6). In terms of collinearity, the tolerance and variance inflation factor (VIF) statistics demonstrated that all influential factors were not strongly correlated. 

### 3.6. Influential Factors on Self-Perceived Dietary Difficulties of Orthodontic Patients

The regression model demonstrated that all independent variables appeared to have no statistically significant impact on the self-perceived dietary difficulties of orthodontic patients (*p* > 0.05) with an R-squared value of 0.075 (Table 7). In terms of collinearity, the tolerance and variance inflation factor (VIF) statistics demonstrated that all influential factors were not strongly correlated. 

## 4. Discussion

The evaluation of the self-perceived treatment needs of orthodontic patients can be considered necessary due to its impact on patient compliance in orthodontic treatment [18]. However, the results of this research demonstrated difficulties in the evaluation of the self-perceived orthodontic treatment needs, in which those perceived by the patients and evaluated by their orthodontic postgraduates were not significantly correlated. There appeared to be a number of physical and psychosocial factors to be considered, meaning that self-perceived needs are sometimes not affected by the severity of malocclusion [19]. There is evidence that more than half of the participants felt satisfied with their dental appearance although the Index of Complexity, Outcome, and Need (ICON) indicated orthodontic treatment needs [20]. On the other hand, patients who are considered by their orthodontists as not having any orthodontic needs may seek treatment, in order to improve their appearances [11,21]. Measurements of OHRQoL should be considered to reinforce clinical assessment in the evaluation of self-perceived treatment needs in orthodontic patients [22]. These arguments support the necessity of OHRQoL questionnaires to assess the treatment needs perceived by orthodontic patients.

Dietary difficulties secondary to orthodontic treatment should also be considered. The findings of this research demonstrate that patients with fixed orthodontic treatment may have dietary difficulties in both the physical and psychological aspects. Patients may have to avoid certain types of food or require longer time for their meal, together with food impaction around their teeth and appliances, which were consistent with a number of studies in this area [10,13,15,16,23]. These limitations seem to be the disadvantages of fixed orthodontic treatment compared to clear aligners [17]. Patients may also feel a lack of confidence or be embarrassed to have a meal with their orthodontic appliance [3]. These difficulties might hinder patient compliance with the treatment. 

Orthodontists are required to evaluate the dietary difficulties of their patients, however, this research identified no significant correlations between the dietary difficulties perceived by patients and those evaluated by their orthodontic postgraduates in both the physical and psychological aspects. Patients with severe malocclusion tend to have negative consequences of OHRQoL, as masticatory limitations may reduce the dietary intake performances [24,25]. However, there appear to be a number of external factors influencing the perceptions toward dietary difficulties apart from malocclusion and dental problems such as the time and location of a meal, and patients with different diet preferences may perceived their dietary difficulties differently [15]. These arguments are supported by the regression models constructed in this research, where the severity and types of malocclusions, together with the professional evaluations, were not significant predictors of self-perceived dietary difficulties. Consequently, there appear to be challenges for orthodontists to provide an accurate evaluation of the dietary difficulties of their patients. 

This research supports the use of patient-reported measurements to assess OHRQoL, in addition to clinical assessment for the evaluation of the treatment needs and dietary difficulties of orthodontic patients. Self-administered questionnaires can be implemented to assess the self-perceived OHRQoL of orthodontic patients [3,14,18,22,26,27,28]. In addition, there is evidence reporting that dental patients do not perceive a self-administered questionnaire as burdensome [29]. These evaluations should therefore be applied before, during, and at the end of orthodontic treatments to identify the physical and psychosocial concerns of patients as well as how they perceive their orthodontic treatment outcomes, rather than solely asking them verbally. Dietary advice should also be provided for orthodontic patients throughout the treatment [30]. Therefore, the OHRQoL measures would allow orthodontic postgraduates to provide specific dietary support or advice to their patients to enhance the treatment outcomes and satisfaction.

This research was conducted in orthodontic residents, which is a uniqueness of this survey. However, there remain the possibility that qualified orthodontists may have similar challenges in evaluating the OHRQoL of their patients. Although orthodontic postgraduates have less experience in orthodontic care compared to qualified orthodontists, they tend to spend more time with their patients in each follow-up appointment compared to qualified orthodontists. Therefore, they could have more chances to realize any OHRQoL concerns of their patients. In other words, the significance of this research could be implemented for both orthodontists and orthodontic residents in the evaluation of OHRQoL perceived by their patients. OHRQoL measures (e.g., a self-administered questionnaire) should be required for them to comprehensively evaluate their patients. There is evidence that quality of life education should be incorporated into the medical and dental curriculum [31,32]. Consequently, the training of OHRQoL measures should be encouraged in orthodontic curricula to enhance the patient-centered care concept for their students and graduates, with the expectation that they would routinely implement this knowledge in their future practice.

The findings of this survey can be applied to other similar contexts even though it was a single-site research, however, the transferability should be confirmed by further studies in other settings in orthodontic education. Likewise, the significance of OHRQoL should be assessed in other disciplines of dental practice. Qualitative research would further enhance an in-depth understanding on how OHRQoL is perceived by patients in orthodontic practice. In addition, future educational research should investigate potential training strategies to encourage orthodontists and orthodontic residents to consider the OHRQoL measures together with a clinical assessment at their dental practice.

## 5. Conclusions

This research identified challenges for orthodontic postgraduates to evaluate the OHRQoL perceived by their patients, especially the self-perceived treatment needs and dietary difficulties. There were also no significant factors to predict the self-perceptions of orthodontic patients in terms of the self-perceived treatment needs and dietary difficulties. Consequently, orthodontic education and practice should encourage the implementation of OHRQoL measures together with clinical examination to assess orthodontic patients, in order to promote the treatment outcomes and satisfaction in accordance with the concept of patient-centered care.

## Figures and Tables

**Table 1 ijerph-20-05446-t001:** Cronbach’s alpha and intraclass correlation (ICC) of each construct in the pilot testing.

Variables	Coefficient Alpha	ICC
**The questionnaire for orthodontic patients**		
Self-perceived treatment needs in functional limitations	0.764	0.863
Self-perceived treatment needs in esthetic problems	0.899	0.917
Self-perceived treatment needs in psychosocial concerns	0.828	0.939
Self-perceived dietary difficulties in physiological limitations	0.795	0.913
Self-perceived dietary difficulties in psychological aspects	0.754	0.904
**The questionnaire for orthodontic postgraduates**		
Evaluated treatment needs in functional limitations	N/A *	0.712
Evaluated treatment needs in esthetic problems	N/A *	0.824
Evaluated treatment needs in psychosocial concerns	N/A *	0.794
Evaluated dietary difficulties in physiological limitations	N/A *	0.786
Evaluated dietary difficulties in psychological aspects	N/A *	0.749

* Note: The coefficient alpha values of the questionnaire for the orthodontic postgraduates are not available as they are not constructs.

**Table 2 ijerph-20-05446-t002:** Cronbach’s alpha of each construct of the PT questionnaire.

Variables	Coefficient Alpha
**The questionnaire for orthodontic patients**	
Self-perceived treatment needs in functional limitations	0.726
Self-perceived treatment needs in esthetic problems	0.886
Self-perceived treatment needs in psychosocial concerns	0.856
Self-perceived dietary difficulties in physiological limitations	0.766
Self-perceived dietary difficulties in psychological aspects	0.802

**Table 3 ijerph-20-05446-t003:** Patient information of the research participants.

Variables		*n*	%
**Angle’s classification**	Class I	59	44.70
	Class II	37	28.03
	Class III	36	27.27
**Vertical dimension**	No problem	64	48.49
	Deep bite	53	40.15
	Open bite	15	11.36
**Transverse dimension**	No problem	103	78.03
	Posterior crossbite	25	18.94
	Scissor bite	4	3.03
**Overjet**	Normal	37	28.03
	Large overjet	56	42.42
	Edge to edge	12	9.09
	Anterior crossbite	27	20.46
**Crowding**	No problem	46	34.85
	Crowding	86	65.15
**Spacing**	No problem	90	68.18
	Spacing	42	31.82
**Treatment phases**	Leveling and aligning	50	37.88
	Space closure	55	41.67
	Finishing	27	20.45

**Table 4 ijerph-20-05446-t004:** OHRQoL perceived by patients and their orthodontists.

Variables	Mean	SD
**The questionnaire for orthodontic patients (5-point Likert scale)**		
Self-perceived treatment needs in functional limitations	2.84	0.76
Self-perceived treatment needs in esthetic problems	3.88	0.93
Self-perceived treatment needs in psychosocial concerns	3.07	0.95
Self-perceived dietary difficulties in physiological limitations	2.96	0.69
Self-perceived dietary difficulties in psychological aspects	2.69	0.75
**The questionnaire for orthodontic postgraduates (100-point VAS scale)**		
Evaluated treatment needs in functional limitations	67.19	24.20
Evaluated treatment needs in esthetic problems	80.60	17.37
Evaluated treatment needs in psychosocial concerns	62.30	27.48
Evaluated dietary difficulties in physiological limitations	60.97	23.57
Evaluated dietary difficulties in psychological aspects	47.58	23.85

**Table 5 ijerph-20-05446-t005:** Correlations of OHRQoL perceived by the patients and evaluated by their orthodontic postgraduates.

Variables	Correlation Coefficient	*p*-Value
**Treatment needs in all aspects**	**0.123**	**0.158**
Treatment needs in functional limitations	0.013	0.882
Treatment needs in esthetic problems	0.157	0.072
Treatment needs in psychosocial concerns	0.126	0.149
**Dietary difficulties in all aspects**	**0.020**	**0.819**
Dietary difficulties during treatment in physiological limitations	0.026	0.768
Dietary difficulties during treatment in psychological aspects	0.024	0.785

**Table 6 ijerph-20-05446-t006:** Multiple regression analysis of the influential factors on the self-perceived treatment needs of orthodontic patients.

Variables	Regression Coefficient	*p*-Value	Tolerance	VIF
Angle’s classification	0.087	0.397	0.492	2.034
Vertical dimension	−0.124	0.186	0.881	1.135
Overjet	0.009	0.921	0.412	2.424
Transverse dimension	−0.013	0.923	0.810	1.234
Crowding	0.026	0.871	0.622	1.608
Spacing	−0.113	0.487	0.621	1.611
Treatment phases	0.077	0.343	0.974	1.026
Treatment needs evaluated by orthodontists	0.005	0.214	0.745	1.342

Note: R-square = 0.063, Significant at *p* < 0.05.

**Table 7 ijerph-20-05446-t007:** Multiple regression analysis of influential factors on the self-perceived dietary difficulties of orthodontic patients.

Variables	Regression Coefficient	*p*-Value	Tolerance	VIF
Angle’s classification	0.018	0.851	0.490	2.040
Vertical dimension	−0.090	0.309	0.881	1.135
Overjet	0.028	0.732	0.405	2.470
Transverse dimension	−0.176	0.174	0.768	1.302
Crowding	0.281	0.067	0.603	1.659
Spacing	0.235	0.130	0.612	1.633
Treatment phases	0.123	0.111	0.973	1.028
Treatment needs evaluated by orthodontists	−0.001	0.806	0.592	1.688
Dietary difficulties evaluated by orthodontists	0.002	0.556	0.695	1.438

Note: R-square = 0.075, Significant at *p* < 0.05.

## Data Availability

The data that support the findings of this study are available from the corresponding author, upon reasonable request. The data are not publicly available due to information that could compromise the privacy of the research participants.

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
