# Peer review of "Significance of Non-Statistically Significant Results in the Prediction of Oral Health-Related Quality of Life in Orthodontic Patients: A Survey Using Two Linking Questionnaires"

_ijerph, 2023, doi:10.3390/ijerph20085446_

Round 1

Reviewer 1 Report

Dear colleagues!

Thank you for the opportunity to review your work, which is devoted to topical issues of prediction of oral health-related quality of life in orthodontic patients.

The main question is about the sample of the number of participants and the formula with which you performed the Sample size calculation.

How were underage patients selected for participation in the study, and were ethics regarding occupational risks observed?

In conclusion, I would suggest adding practical recommendations for doctors

With regard to the list of references that the authors use, I want to draw attention to a sufficient number of references that are more than 10 years old: 1, 6, 7, 12, 18, 19, 20, 31, 32.

In table 3, do the percentages reflect the number of total participants or within each group? If so, what should be recalculated taking into account 100% within each block or indicated in the comments.

The topic of the study is relevant in the field, however, it is necessary to describe specific practical recommendations for practicing physicians who can use the results of the study in their work.

Author Response

Dear Reviewer,

We do really appreciate your helpful suggestions on improving our research article. We have revised our paper following your advice. The details of the revisions are provided in the table ‘Author’s Reply to the Review Report’.

Thank you again for your valuable support. If you have any further suggestions, could you please let us know?

Sincerely yours,

Kawin Sipiyaruk

Reviewer 2 Report

1. What is the translational aspect of this scientific research?

2. What is the future continuation of this survey?

3. What is the limitation of your research?

Author Response

(The authors gave the same response as above.)

Reviewer 3 Report

The paper by Santiwong et al. is about a Survey focused on a PROM outcome as OHRQoL in orthodontic patients. The study has provided a sound example of a Survey build up, piloting and validation and its application to a group of Ortho patients. Athough non statistical significative results have been provided, there is an evidence for a qualitative analysis of the outcomes. It would be interesting to have information about results stratified according to different variables as: age ( or groups according to age spans), different types of ortho treatments/devices and treatment length, to check their effects on the considered aspects (aesthetics, function and psychological impact). Other than that the paper is eligible for publication after minor revisions.

Author Response

(The authors gave the same response as above.)

Round 2

Reviewer 1 Report

Dear authors!

Thank you for your work. I am satisfied with the results and have no other questions.